# CO$_2$-promoted photocatalytic aryl migration from nitrogen to carbon for switchable transformation of *N*-arylpropiolamides

Ge Liu[1,2,3,4], Denghui Ma[2], Jianchen Zhang[1,3,4], Fanyuanhang Yang[1,3,4], Yuzhen Gao [1,3,4] ✉ & Weiping Su [1,3,4] ✉

Photocatalytic N-to-C aryl migration allows for quick construction of highly useful amide derivatives from readily available compounds. By developing the reactions of sodium sulfinates with the *N*-aryl-propiolamides, we herein demonstrate that the CO$_2$-promoted visible-light-induced method enables a large variety of aryl groups on nitrogen atoms of the *N*-arylamides to undergo efficient aryl migration from N atom to C atom to synthesize *tetra-* and *tri*-substituted alkenyl amides selectively. 1,4-N-to-C aryl migration is a key step in this transformation which is achieved through photocatalytic radical-polar crossover pathway. The protocol exhibits the remarkably tolerant of the electronic properties of the migrating aryl substituent, as both electron-rich and -poor arenes are compatible with the migration process. As a result, this protocol features with a broad substrate scope, as demonstrated by more than 90 examples including complex bioactive compounds. Notably, abundant, nontoxic and low-cost CO$_2$ acted as an essential and irreplaceable additive to enable the *tetra-* and *tri*-substituted alkenyl amides to be synthesized with excellent selectivity.

The visible-light-induced radical aryl migration reaction, namely radical Truce−Smiles rearrangement, is one of the most powerful transformations to synthesize challenging compounds from the readily available substrates[1,2]. These protocols avoid using toxic reagents, stoichiometric oxidants, or strong bases and feature the simple reaction conditions for intramolecular migration of aryl groups. Therefore, the methods for visible-light-induced radical aryl migrations have been intensively studied and applied in syntheses, and a large variety of functional molecules have been constructed through these approaches[3–5]. Amide is a ubiquitous and important core in many natural products and synthetic drugs[6,7]. Consequently, there is a spontaneously increased demand for the synthesis of structurally diverse amide derivatives via more efficient and sustainable pathways. Undoubtedly, the radical Truce−Smiles rearrangement is an appealing

and concise approach to constructing structurally diverse amides, considering the uniqueness of such a transformation mode. Surprisingly, although radical Truce−Smiles rearrangement based on the cleavage of N−SO$_2$Ar bond by following the extrusion of SO$_2$ for the synthesis of amide derivatives have been well developed[1,2,8–11], the examples of visible-light-induced radical aryl migration from N atom to C atom are sparse[12–17]. In this regard, the Stephenson's group[12] and Wang's group[13] independently reported photo-induced *intra-* or *inter*-molecular reactions of *N*-(hetero)arylamines resulting in 1,4-N-to-C radical aryl migration by homolytic C−N bond cleavage (Fig. 1a, eq. 1). However, these elegant migrations via Truce−Smiles rearrangement are strongly rely on the electronic nature of the migratory aromatic ring, as the migratory arenes were limited mostly to thiophenes or few electron-rich benzene derivatives. Very recently, Greaney and

[1]State Key Laboratory of Structural Chemistry, Fujian Institute of Research on the Structure of Matter, Chinese Academy of Sciences, Fuzhou 350002, PR China. [2]School of New Energy, Ningbo University of Technology, Ningbo 315336, PR China. [3]Fujian Science & Technology Innovation Laboratory for Optoelectronic Information of China, Fuzhou, Fujian, PR China. [4]University of Chinese Academy of Sciences, Beijing 100049, PR China. ✉e-mail: gyz@fjirsm.ac.cn; wpsu@fjirsm.ac.cn

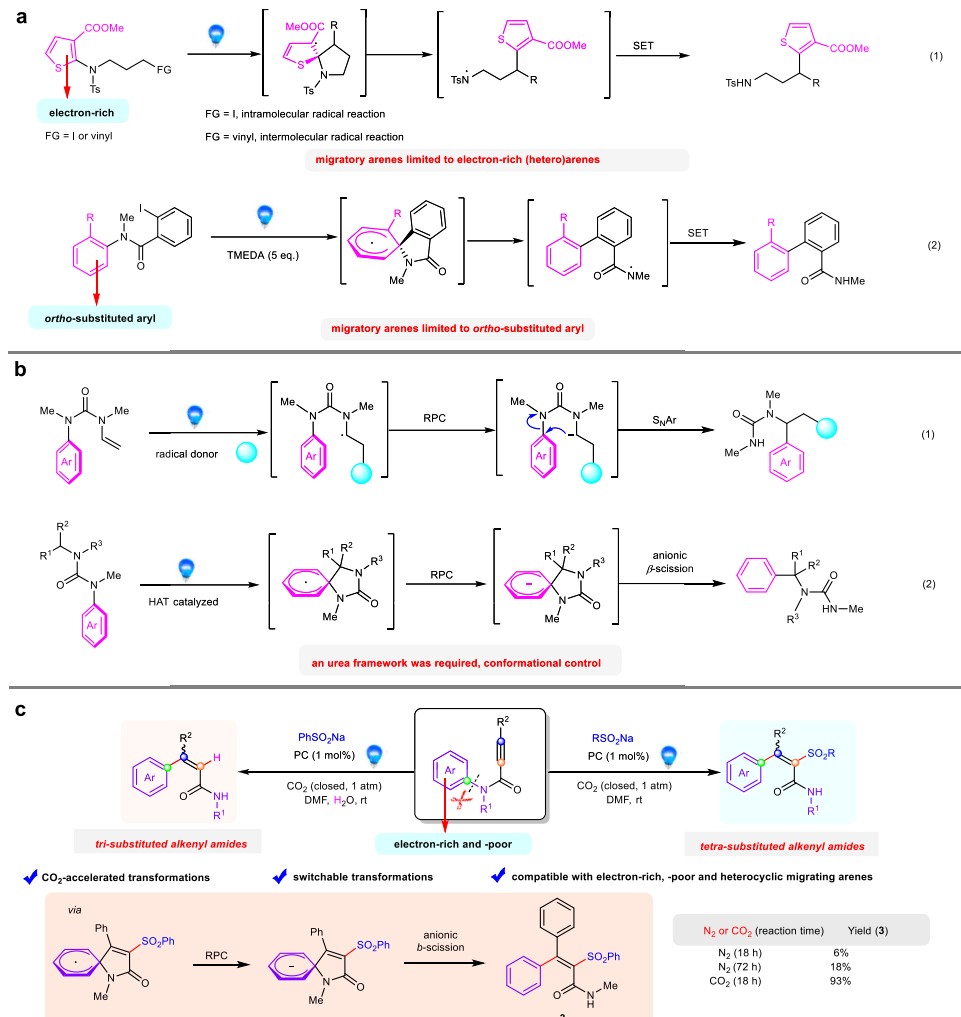

**Fig. 1 | Photo-catalyzed N to C aryl migration reactions. a** Aryl migration from N to C via photocatalytic homolytic C−N bond cleavage. **b** Aryl migration from N to C via photocatalytic radical-polar crossover. **c** This work: CO₂-Promoted photocatalytic radical aryl migration from N to C: switchable synthesis of *tetra*- and *tri*-substituted alkenyl amides. FG functional group, RPC radical-polar crossover, HAT hydrogen atom transfer.

co-authors disclosed the synthesis of biaryl motif from iodoamide substrates by photocatalytic homolytic cleavage of C−N bonds, yet the migratory arenes were limited mostly to *ortho*-substituted benzene derivatives (Fig. 1a, eq. 2)[14]. The intramolecular nucleophilic aromatic substitution reactions (SₙAr substitution) via photocatalytic radical-polar crossover (RPC) pathway resulting in N-to-C aryl migration was well developed by Clayden's group (Fig. 1b, eq. 1)[15], thus offering an important alternative to achieve 1,4-N-to-C aryl migration. During the revision of this manuscript, a similar strategy was reported by Jiang and co-authors, which was triggered by photoredox-mediated hydrogen atom transfer (HAT) (Fig. 1b, eq. 2)[16]. Notably, a urea framework was required for the success of these photocatalytic RPC Truce−Smiles rearrangements (Fig. 1b), which might result from the conformational rigidity of substrates[18,19]. As a result, developing a general and practical method to synthesize biologically important amide derivatives from a wide range of simple precursors via photo-catalyzed radical Truce−Smiles rearrangement through the cleavage of C−N(aryl) bond is still highly desirable in spite of its high challenge.

Polysubstituted olefins are commonly found in pharmaceuticals, natural products, polymers, and other classes of compounds[20,21]. However, the synthesis of polysubstituted olefins, particularly all-carbon *tetra*-substituted alkenes, is of great challenge due to the significant steric hindrance[22–24]. The introduction of an amide group into polysubstituted olefins provides a possibility to further improve their

biological and material properties. Herein, we demonstrate that CO₂-promoted visible-light-induced aryl migration enabled the electronically diverse aryl groups on nitrogen atoms to undergo efficient N-to-C aryl migration. The reaction between sodium sulfinates and *N*-aryl-lamides produced valuable *tetra*- and *tri*-substituted alkenyl amides selectively, thus offering a concise and efficient method to synthesize structurally diverse amide derivatives (Fig. 1c). 1,4-N-to-C aryl migration is a key step in this transformation, which is achieved through photocatalytic RPC pathway. Interestingly, these photocatalytic reactions were greatly accelerated by CO₂ gas (93% yield) compared with the reactions occurring in the absence of CO₂ gas (<20% yield), which disclosed that the CO₂ additive played a key role in promoting this photocatalytic N-to-C radical aryl migration reaction.

## Results
### Reaction condition optimization
At the beginning of the investigations, *N*-methyl-*N*,3-diphenylpropio-lamide (**1a**) and sodium benzenesulfinate (**2a**) were employed as the model substrates. As shown in Table 1, after extensive examination of a range of parameters, such as photocatalysts (PCs) and solvents, the targeted radical-aryl-migration product, namely *tetra*-substituted alkenyl amide (**3**) was obtained in an excellent isolated yield (93%) by using Ir[(dFCF₃ppy)₂(bpy)]PF₆ as the PC under 1 atm of CO₂ with the irradiation of 40 W blue LEDs (entry 1). Notably, when the process was

## Table 1 | Optimization of reaction conditions

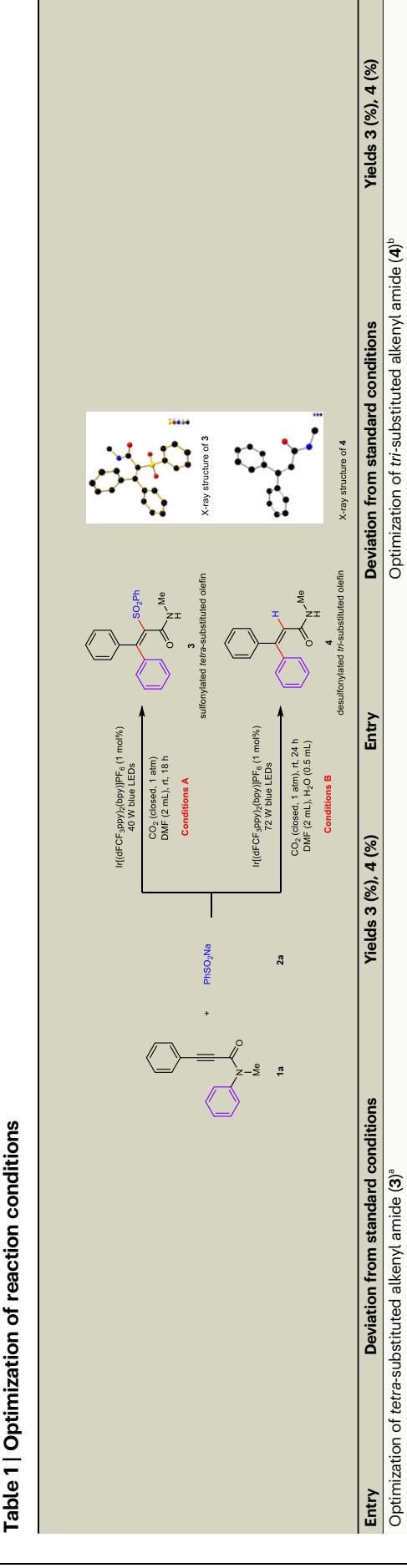

| Entry | Deviation from standard conditions | Yields 3 (%), 4 (%) | Entry | Deviation from standard conditions | Yields 3 (%), 4 (%) |
|---|---|---|---|---|---|
| | Optimization of *tetra*-substituted alkenyl amide (**3**)[a] | | | Optimization of *tri*-substituted alkenyl amide (**4**)[b] | |
| 1 | Conditions A | 95 (93)[c], 0 | 10 | Conditions B | 0, 85 (76)[c] |
| 2 | Under $N_2$ | 6, 0 | 11 | Under $N_2$ | 34, 27 |
| 3 | w/o Ir[(dFCF$_3$ppy)$_2$(bpy)]PF$_6$ | 0, 0 | 12 | w/o Ir[(dFCF$_3$ppy)$_2$(bpy)]PF$_6$ | 0, 0 |
| 4 | w/o light | 0, 0 | 13 | w/o light | 0, 0 |
| 5 | DMSO instead of DMF | 86, 0 | 14 | PhSO$_2$Na (1.0 equiv) | 25, 51 |
| 6 | CH$_3$CN instead of DMF | 80, 3 | 15 | PhSO$_2$Na (0.5 equiv) | 40, 5 |
| 7 | 4CzIPN as PC | 85, trace | 16 | AcOH (5.0 equiv) instead of H$_2$O | 32, 16 |
| 8 | Ir(ppy)$_2$(dtbbpy)·PF$_6$ as PC | 47, 27 | 17 | 4-CF$_3$PhSO$_2$Na instead of PhSO$_2$Na | 0, 64 |
| 9 | Ru(bpy)$_3$Cl$_2$ as PC | 79, 7 | 18 | 2-OMePhSO$_2$Na instead of PhSO$_2$Na | 76[d], 0 |

[a]Reaction conditions: **1a** (0.1 mmol), **2a** (0.15 mmol, 1.5 equiv), Ir[(dFCF$_3$ppy)$_2$(bpy)]PF$_6$ (1 mol%), DMF (2 mL), 1 atm CO$_2$, 40 W blue LEDs, rt, 18 h (conditions A). Yield was determined by $^1$H NMR with CHCl$_2$CHCl$_2$ as internal standard. [b]Reaction conditions: **1a** (0.1 mmol), **2a** (0.15 mmol, 1.5 equiv), Ir[(dFCF$_3$ppy)$_2$(bpy)]PF$_6$ (1 mol%), H$_2$O (0.5 mL), DMF (2 mL), 1 atm CO$_2$, 72 W blue LEDs, rt, 24 h (conditions B). Yield was determined by $^1$H NMR with CHCl$_2$CHCl$_2$ as internal standard. [c]Yield of isolated product shown in parentheses. [d]The product number is **50**. dFCF$_3$ppy 2-(2,4-difluorophenyl)–5-(trifluoromethyl)pyridine, dtbbpy 4,4-di-*tert*-butyl-2,2′-bipyridine, ppy 2-phenylpyridine; bpy 2,2′-bipyridine; dtbbpy 4,4-di-*tert*-butyl-2,2′-bipyridine, 4CzIPN 2,4,5,6-*tetra*(9*H*-carbazol-9-yl)isophthalonitrile.

performed under an $N_2$ atmosphere without $CO_2$, a low yield (6%) was detected, and most of **1a** remained unreacted (entry 2), indicating the essential role of $CO_2$ in this reaction. Further control reactions revealed that photocatalyst (PC) and light were both vital to this transformation, as none of the expected products was detected when the reaction was conducted in the absence of one of them (entries 3 and 4). Some other solvents, such as DMSO and $CH_3CN$, also afforded the desired product in a slightly decreased yield (entries 5 and 6). The explorations of other PCs showed that 4CzIPN gave a comparable yield of **3** (entry 7). [Ir(ppy)$_2$(dtbbpy)](PF$_6$) and Ru(bpy)$_3$Cl$_2$ led to a decrease in the yield of **3**, because product **3** underwent desulfonylation to produce *tri*-substituted alkenyl amide (**4**) (entries 8 and 9). The detection of product **4** encouraged us to further explore the reaction conditions to efficiently synthesize product **4** and its analogs. Pleasingly, it was found that in the presence of $CO_2$ and an appropriate amount of $H_2O$, this desulfonylated olefin (**4**) was isolated in 76% yield as a single product when the reaction was performed under the irradiation of 72 W blue LEDs by prolonging the reaction time to 24 hours (entry 10). Mixed products were afforded when $CO_2$ was replaced by $N_2$ in this desulfonylation reaction (entry 11), indicating that $H_2O$ played a role similar to $CO_2$ under photoredox catalytic conditions (entry 2 vs entry 11)[25], but was less efficient than $CO_2$ in controlling chemoselectivity of reactions. The PC and light were both essential for the desulfonylated process (entries 12 and 13). Moreover, the yield of **4** was decreased sharply, and mixed products (**3** and **4**) were obtained when the loading of sodium benzenesulfinate (**2a**) was reduced to 1.0 or 0.5 equivalent (entries 14 and 15), indicating the sodium benzenesulfinate did not act as a catalyst in this desulfonylated process. The $H_2O$ could not be replaced by HOAc, since HOAc offered mixed products (entry 16). Interestingly, when 4-CF$_3$PhSO$_2$Na was employed in place of **2a**, the target desulfonylated product **4** was also produced in a slightly decreased yield (entry 17). In contrast, 2-OMePhSO$_2$Na gave the sulfonylated product **50** (see Fig. 2) in high yield without formation of the desulfonylated product **4** (entry 18). These two different results (entry 17 vs entry 18) might be attributed to the fact that electron-rich aryl sulfinates are poorer leaving groups than electron-poor ones[26,27].

## Substrate scopes

With the optimal conditions in hand, then a broad range of propiolamides possessing electronically diverse groups was tested to investigate the generality of the N-to-C radical aryl migration process. As shown in Fig. 2, substrates bearing alkyl *N*-protecting groups (such as *N*-methyl, *N*-ethyl, *N*-benzyl, and *N*-allyl) were initially examined, and all offered the desired *tetra*-substituted alkenyl amides in generally good yields (**3, 5–8**). Importantly, the *N*-allyl group, which is considered to be an ideal radical acceptor, was also compatible with this migration process (**6**), albeit in a relatively low yield. This may be due to sulfonyl radical acting as an effective leaving group, thus regenerating the allyl group followed by β-elimination of the sulfonyl radical[28]. In the case of employing secondary propiolamide as the substrate (with *N*−H), the desired product **9** was also isolated in high yields. Moreover, *N*,*N*-diaryl propiolamides were also compatible with the migratory transformation (**10-12**), and the aryl migratory ability of different benzyl tethered to N atom was also examined. Both benzyl and 4-fluorobenzyl groups were able to serve as migratory arenes, but the relatively more electron-poor 4-fluorophenyl group was preferred, resulting in a regioselective ratio (rr) of 3:1 (**11**). Another pair comprising 4-methoxybenzyl and benzyl groups showed lower selectivity to produce **12** with rr = 1.4:1. Electron-donating (such as methyl, methoxy, phenyl and hydroxy) and electron-withdrawing (such as trifluoromethyl and trifluoromethoxy) substituents in the *para*- and *meta*-position of the aryl ring attaching to alkynyl carbon atoms were well tolerated to produce the expected products (**13-19**) in good yields. Moreover, the propiolamides tethered with naphthalene and biologically important hetero-arenes (such as furan, indazole, dibenzofuran,

dibenzothiophene, and Ketoprofen derivatives) on the aryl ring attaching to alkyne group were all well tolerated to produce the corresponding products (**20-26**) in generally good yields. With the broad scope of the different *N*-protecting groups and alkyne aryl rings established, variation in the migratory aryl (*N*-aryl ring) substituent was examined subsequently. Notably, a wide range of *N*-aryl groups possessing electron-donating groups (such as methyl, methoxy, phenyl) and electron-withdrawing groups (such as trifluoromethyl and ester) smoothly underwent N-to-C migration to deliver the corresponding alkenyl-sulfone products (**27-29, 33-35**) in generally good yields, which demonstrated the good generality of this N-to-C aryl migration process with regard to the migrating aryl groups. The halogenated aryl groups (fluoro, chloro, bromo, and iodo) bound to nitrogen atoms of propiolamides all acted as the migrating groups to afford the target product products in good to excellent yields with halogen-groups untouched (**30-32, 36-38**), thus providing an ample opportunity for further elaboration of the products generated from this photoredox catalyzed radical aryl migratory reactions. Notably, the steric hindrance has no influence on the rearrangement, as was shown by the propionamide bearing *ortho*-substituted aryl groups on nitrogen atoms that offered the target products **35** and **36** in excellent yields. Finally, naphthyl, benzo-1,3- dioxol, dibenzofuryl, and thienyl substitutions did not hamper the aryl migration process, to offer products (**39-42**) in acceptable yields. Unfortunately, the attempt of 1,5-C to C aryl migration failed to occur, and alkene amide or β-methyl propiolamide also could not furnish the desired products. In addition, sulfonyl radical was more likely to attack the terminal position of alkyne when β-unsubstituted propiolamide was employed as the substrate to yield **43** in 30% yield.

Then we turned our attention to explore the scope of the sulfinates. A variety of aryl sulfinates with electron-donating groups (such as methyl, *iso*-propyl, and methoxy) or electron-withdrawing groups (fluoro, chloro, bromo, trifluoromethyl, ester, cyano, and acetyl) on the aryl rings reacted efficiently with **1a** to give diverse alkenyl sulfones (**44-55**) in generally good yields, indicating the electronic properties of the aryl rings of sulfinates had little effect on this transformation. In addition, naphthylsulfinates were viable for this radical aryl migration in good yields (**56** and **57**). Remarkably, sulfinates possessing biologically important hetero-arenes, such as pyridine (**58**), quinolone (**59**) and thiophene (**60**), were all successfully subjected to this transformation and resulted in the formation of corresponding heterocyclic alkenyl sulfones in acceptable yields. Moreover, aliphatic sulfinate, such as sodium methylsulfinate, also served as a suitable coupling partner for this transformation **61**. Notably, sulfinates derived from Valdecoxib and Sildenafil were both compatible well with the transformation (**62** and **63**).

Traditionally, the sulfonyl moiety is usually removed through a reductive-desulfonylation process, thus an external reductant is required in the transformation[28–31]. As shown in Table 1, this photocatalytic conditions, in combination with $H_2O$ additive, enabled *tetra*-substituted alkenyl amides to undergo desulfonylation in the absence of any reductant for the generation of *tri*-substituted alkenyl amides. Undoubtedly, this is a mild and direct approach to preparing *tri*-substituted alkenyl amides. Therefore, it's meaningful to expand the substrate scope to synthesize diverse *tri*-substituted alkenyl amides from easily available starting materials under mild conditions. As shown in Fig. 3, when propiolamides possessing electron-donating and -withdrawing substituents both in the alkyne aryl ring (**4, 64-77**) and the migratory aryl (*N*-aryl ring) (**78-90**) were employed as substrates, the reactions progressed competently delivering the corresponding *tri*-substituted alkenyl amides in generally good yields. Propiolamides derived from Ketoprofen, Probenecid, and Estrone could all be transformed into the targeted products (**75-77**) in moderate to good yields. Moreover, propiolamides tethered with hetero-arenes to the alkyne-(**73, 74, 90**) or *N*-positions (**89**) were all successfully subjected to this

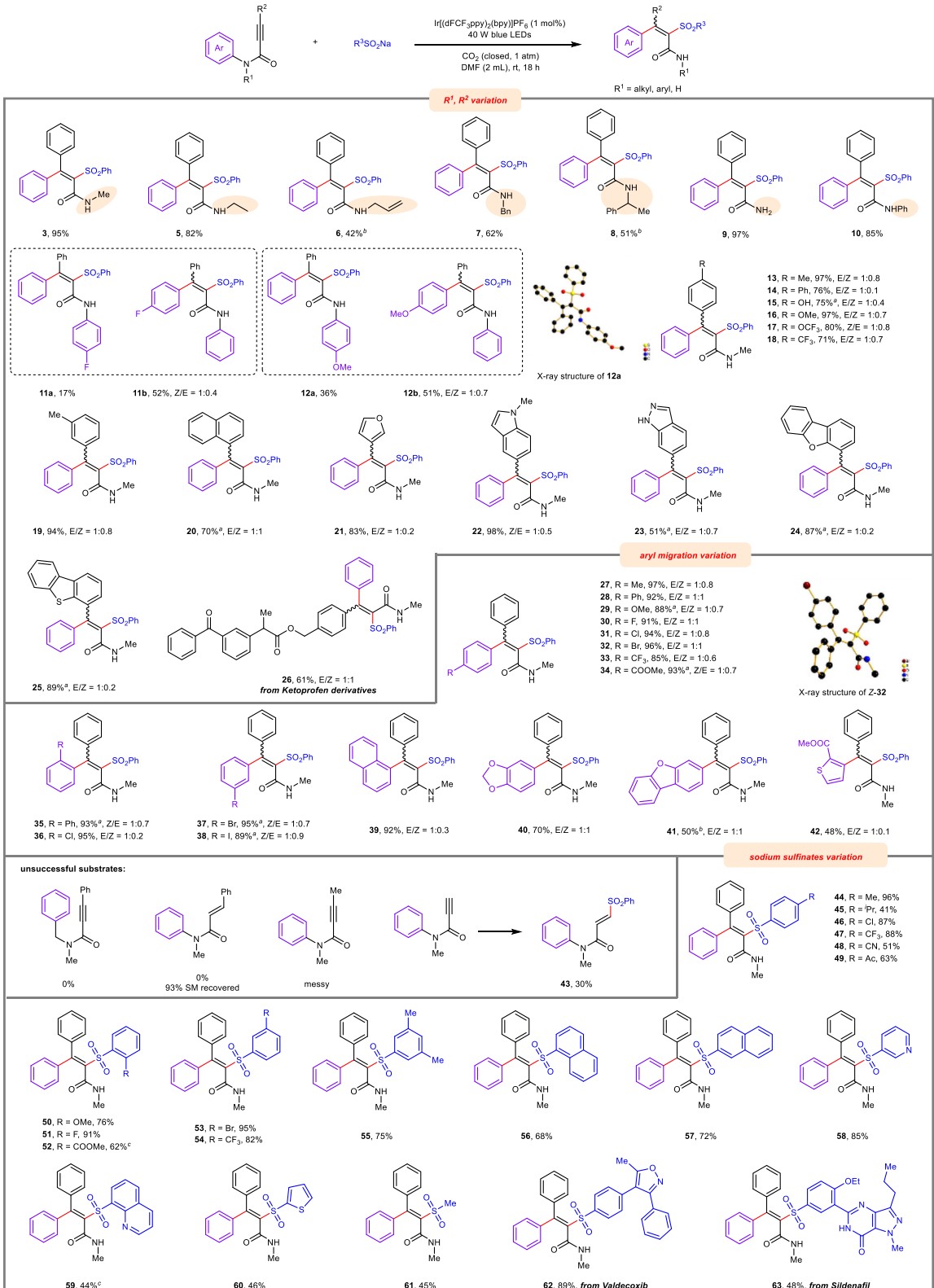

**Fig. 2 | Substrate scope of the method for formation of *tetra*-substituted alkenyl amides.** Reaction conditions: **1** (0.1 mmol), $R^3SO_2Na$ (0.15 mmol, 1.5 equiv), Ir[(dFCF$_3$ppy)$_2$(bpy)]PF$_6$ (1 mol%), DMF (2 mL), 1 atm $CO_2$, 40 W blue LEDs, rt, 18 h. Isolated yields. [a]24 h. [b]48 h. [c]H$_2$O (50 μL) was added.

tandem radical aryl migration/desulfonylation transformation to give the corresponding products in reasonable yields. Halogenated *N*-aryl groups (fluoro, chloro, bromo, and iodo) were all viable in the target transformation and afforded the desired halogenated *tri*-substituted alkenyl amides in good yields (**81-83, 86-88, 90**).

Moreover, with a slight modification of the reaction conditions, we found that sodium trifluoromethanesulfinate ($CF_3SO_2Na$) was also a successful precursor, furnishing the corresponding $CF_3$-containing *tetra*-substituted alkenyl amides (**91-95**) in acceptable yields with good Z/E selectivity via a cascade process involving trifluoromethyl radical

**Fig. 3 | Substrate scope of the method for the formation of *tri*-substituted alkenyl amides.** Reaction conditions: **1** (0.1 mmol), PhSO$_2$Na (0.15 mmol, 1.5 equiv), Ir[(dFCF$_3$ppy)$_2$(bpy)]PF$_6$ (1 mol%), DMF (2 mL), H$_2$O (0.5 mL), 1 atm CO$_2$, 72 W blue LEDs, rt, 24 h. Isolated yields. [a]48 h. [b]H$_2$O (1 mL), 48 h.

**Fig. 4 | Scope of propiolamides in the trifluoromethyl-arylation reaction.** Reaction conditions: **1** (0.2 mmol), CF$_3$SO$_2$Na (0.4 mmol, 2.0 equiv), 4CzIPN (2 mol%), K$_2$CO$_3$ (0.3 mmol, 1.5 equiv), DMSO (2 mL), 40 W blue LEDs, N$_2$, rt, 24 h.

addition to the C−C triple bond and C−N bond cleavage concomitant remote aryl migration (Fig. 4). Notably, this transformation was achieved successfully in the absence of CO$_2$.

## Synthetic applications

Photocatalytic isomerization of the readily available E/Z mixtures allows direct access to significantly important multiple substituted olefins with high stereoselectivity[32,33]. After careful investigation of photocatalysis, solvent, and light source, the *tetra*- and *tri*-substituted alkenyl amides with good Z/E ratio were successfully accessed by using DCA (anthracene-9,10-dicarbonitrile) as the photocatalysis in EtOAc irradiated under blue LEDs (Fig. 5a). However, it fails to further increase the Z/E ratio of the *tetra*-substituted alkenyl amide for which acceptable Z/E ratios have been obtained in Fig. 2 (such as **21, 24, 25,** and **36**). To the best of our knowledge, the photocatalytic isomerization of *tetra*-substituted olefins has not been achieved until now, presumably due to the Z and E isomers of these *tetra*-substituted olefins do not possess sufficient difference in their respective triplet

energies[32,33]. Notably, a one-pot, two-step strategy was developed for the concise syntheses of *tetra*- or *tri*-substituted alkenyl amides directly from easily available arylamines in acceptable yields (Fig. 5b). Moreover, this protocol enabled the gram-scale reactions with slight decrease in the yields of **3** and **30** when the loading of PC was reduced to 0.1 or 0.5 mol% (Fig. 5c), demonstrating the good practicability of this protocol.

## Mechanism investigations

Preliminary studies were carried out to probe the mechanism of this reaction. Initially, Stern−Volmer luminescence studies revealed that the excited state of photo-catalyst (PC*) was more likely to be quenched by PhSO$_2$Na than substrate **1a** (see Supporting Information for details). When the model reaction was conducted in the presence of 2,2,6,6-tetramethylpiperidin-1-oxyl (TEMPO), quinone (BQ) or ethene-1,1-diyldibenzene, the reaction was almost suppressed, suggesting that a radical pathway may involve in this transformation (Fig. 6a). This inference was further supported by the experiment of using "radical

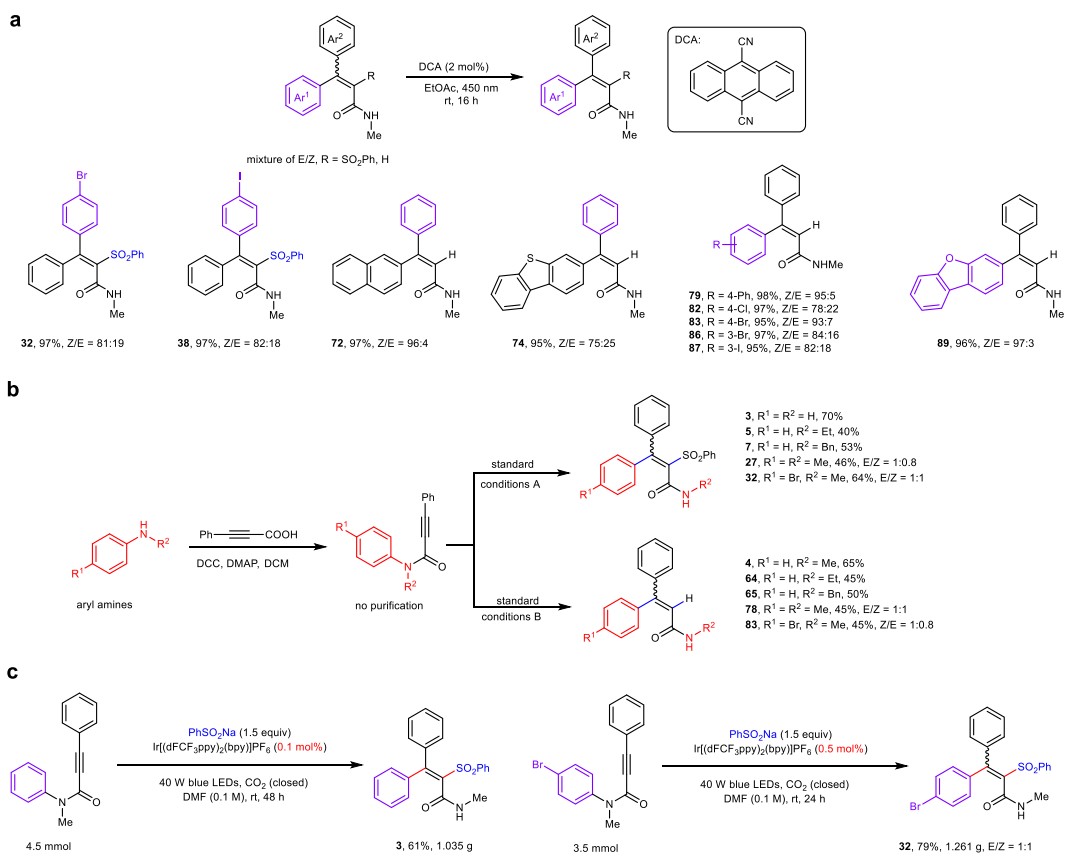

**Fig. 5 | Synthetic applications. a** Photocatalytic isomerization of the *tetra*- and *tri*-substituted alkenyl amides. **b** One-pot two-step synthesis of *tetra*- and *tri*-substituted alkenyl amides. **c** Gram-scale experiments.

clock" substrate **96** in which the ring-opening product **97** was isolated (Fig. 6b). The hydrogen atom bonded to nitrogen (N–H) in the product **3** was not deuterated when the reaction was conducted in $d_6$-DMSO, ruling out the possibility of hydrogen abstraction process (HA) from the solvents (Fig. 6c, eq. (1))[34–36]. Additionally, when $H_2O$ was replaced by $D_2O$, deuterium product *d*−**4** was isolated with high vinyl D/H ratio, suggesting the possible presence of an anion intermediate in the desulfonylated process. The sulfonyl-added/protonated product **98** was not detected, excluding the possibility of intramolecular nucleophilic aromatic ($S_NAr$) *ipso*-substitution (Fig. 6c, eq. (2))[15,18]. Control experiment revealed that the *tetra*-substituted alkenyl amide **3** was able to deliver desulfonylated product **4** under photoredox catalytic in the absence of $CO_2$. On the contrary, when the transformation was conducted without PC, light or $H_2O$, product **3** could not undergo desulfonylation successfully (Fig. 6d, eq. (1)). Moreover, Stern−Volmer luminescence experiment showed that the excited state of photocatalyst (PC*) also could be quenched by product **3** (see Supporting Information for details). These results indicated that the desulfonylated products are produced from *tetra*-substituted alkenyl amides via photoredox catalytic pathway in the presence of enough amount of water. To further explain this desulfonylation process, EtOH was used as the hydrogen source instead of $H_2O$ in this desulfurization reaction, oxidized compound **99**[37–40] was detected by GC-Ms which further indicates that this desulfurization might be through a SET process (Fig. 6d, eq. (2)). Theoretically, the cleavage of C−N bond in the spiro-cyclic intermediate ensure the certain configuration of the products, yet efficient Z/E isomerization of pure *E*-**30** product was observed merely under the irradiation of light (Fig. 6e). These results indicated that the *tetra*-substituted alkenyl amides are able to go through the photochemical Z/E isomerization process, thus explaining the poor Z/E selectivity of products[32,33,41]. When *N*-arylsulfonyl imide was

employed as the substrate, the expected 1,5-aryl-shifted product was also obtained in an excellent yield via desulfonylation process under $CO_2$. On the contrary, when the reaction was conducted under $N_2$, the transformation was almost restrained (Fig. 6f). In addition, it was found that the single-electron reduction of sulfonyl-contained intermediates to generate anion intermediates required the existence of an acid[42–44]. Therefore, some control experiments were performed by adding acids to the reaction system in the absence of $CO_2$ to explore the role of $CO_2$ in this reaction. As the results shown in Fig. 6g, the products could also be obtained in the presence of Bronsted acid, yet in low yields or with low selectivity. Oppositely, Lewis acid, such as $ZnF_2$, could not access satisfactory results. As a result, we assumed that the role of $CO_2$ might be similar to that of Bronsted acids[45,46]. Notably, $CO_2$ is able to synthesize *tetra*- and *tri*-substituted alkenyl amides with excellent selectivity in generally good yields. We believed that the protonic acid could not only promote the formation of *tetra*-substituted alkenyl amides, but also facilitate the generation of *tri*-substituted alkenyl amides (Figs. 3 and 6d), thus yielding the bad selectivity of products. These results demonstrate that $CO_2$ is essential and irreplaceable for achieving the target products with excellent selectivity.

On the basis of these mechanistic studies, a tentative pathway for the $CO_2$-promoted photocatalytic reaction of *N*-aryl-propiolamides with sulfinates via radical aryl migration has been proposed (Fig. 7a). Firstly, blue light irradiation on PC gives rise to the excited PC* [$E_{red}$ (PC*/PC$^{•-}$) = +1.32 V *vs* SCE for Ir[(dFCF$_3$ppy)$_2$(bpy)]PF$_6$][47], which oxidizes the sulfinate anion (**A**) (−0.37 V *vs* SCE for PhSO$_2$Na)[48] via a single-electron transfer (SET) to give the sulfonyl radical (**B**) as well as the anionic intermediate (PC$^{•-}$) of photocatalyst. The regioselective attack of the electrophilic sulfonyl radical **B** to the α carbon atom of C≡C triple bond in substrate propiolamides **1** generates alkenyl radical **C**, which undergoes a

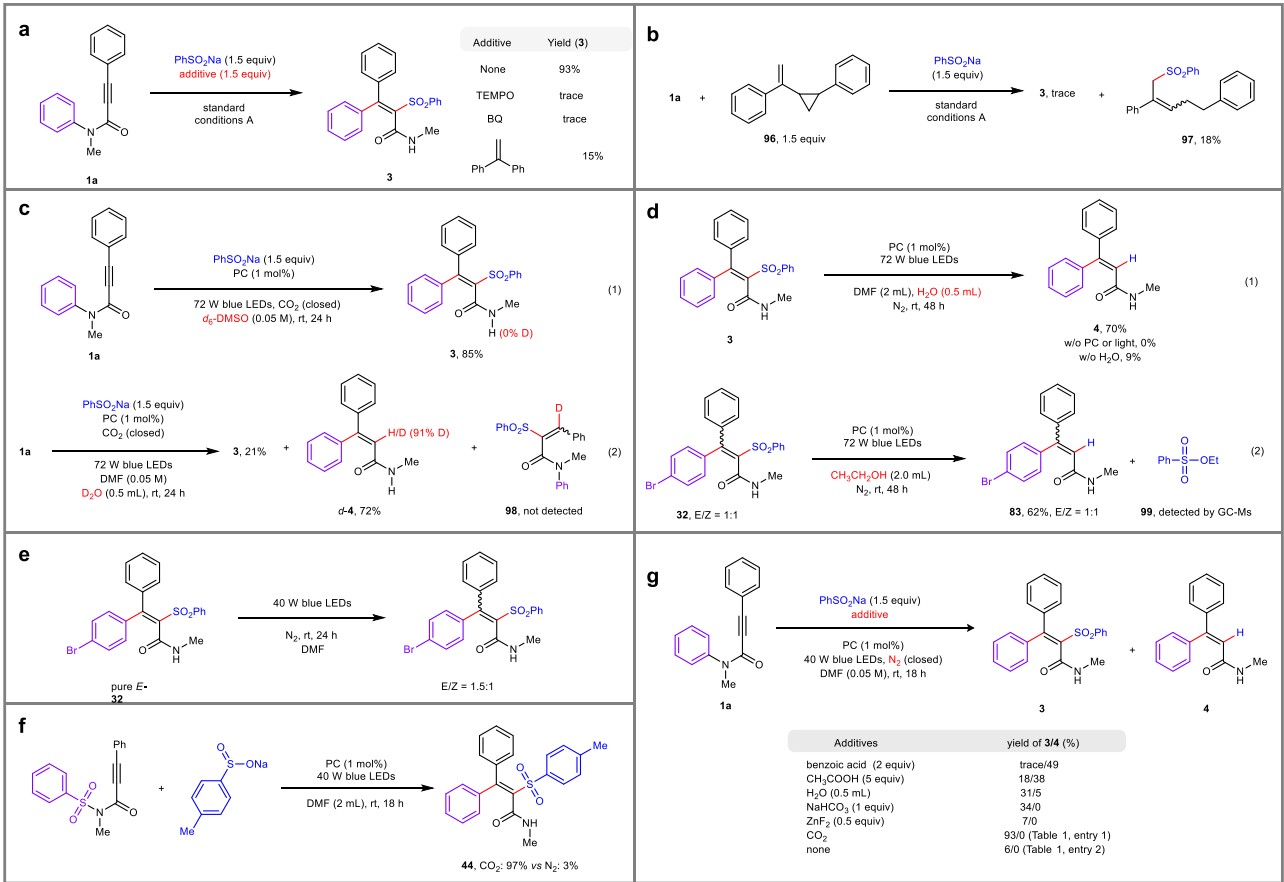

**Fig. 6 | Mechanistic studies. a** Control experiment with radical scavengers. **b** Reaction with radical clock. **c** Isotope-labeling experiments. **d** Desulfonylated exploration. **e** Z/E isomerization in the presence of photocatalysis. **f** Transformation using *N*-arylsulfonyl amide under the standard reaction conditions. **g** Investigation of the role of $CO_2$ in this transformation. PC: Ir[(dFCF_3ppy)_2(bpy)]PF_6.

radical *ipso*-addition[49–51] to the benzene core to produce a spirocyclic radical **D**. DFT calculations was performed in order to provide the energy landscape of the key intermediates (Fig. 7b). It was found that the cleavage of the C−N bond via a spirocyclic anionic pathway was quite facile, associated with a small energetic barrier at the transition state **TS_E** (1.9 kcal mol$^{-1}$, blue line). By comparison, the homolytic cleavage of the C−N bond via a radical process presented significant challenges, with a high energetic barrier at the transition state **TS_D** (17.6 kcal mol$^{-1}$, black line). In addition, the single-electron reduction from **D** to **E** was feasible ($E_{1/2}°$ = −1.17 V vs SCE; calculation details see SI) in the presence of Ir[(dFCF_3ppy)_2(bpy)]PF_6 [$E_{1/2}^{red}$ (PC/PC$^{•−}$) = −1.37 V *vs* SCE for Ir[(dFCF_3ppy)_2(bpy)]PF_6] (Fig. 7c)[47]. Therefore, radical **D** is reduced by PC$^{•−}$ to generate the spirocyclic anionic intermediate **E** which undergo a β-scission process to finish the 1,4-aryl migration process to form the target sulfonylated products **F**. The residual crystal water from the sodium sulfonates might account for the source of hydrogen atoms in product **F**. The Z/E isomerization of **F** is possible through a photo-induced energy transfer manifold to afford **F′**[41]. Besides, in the presence of photocatalyst and enough amount of $H_2O$, *tetra*-substituted alkenyl amides (**F** and **F′**) undergoes photo-driven desulfonylation to produce *tri*-substituted alkenyl amides (**I**) together with the release of RS(O)_2OH. Initially, both **F** and **F′** were reduced by the excited photocatalyst PC* to afford the PC$^{•+}$ [$E_{ox}$ (PC*/PC$^{•+}$) = −1.00 V vs SCE for Ir[(dFCF_3ppy)_2(bpy)]PF_6][47] and corresponding radical anion intermediate **G** ([$E_{red}$ = −1.18 V vs SCE for compound **3**; calculation details see SI). Then, the radical anion **G** undergoes protonation with $H_2O$ to generate the radical intermediate **H** that undergoes β-fragmentation[28] to form the *tri*-substituted alkenyl amides **I** and

sulfonyl radical **B**. Finally, the released sulfonyl radical **B** is oxidized by PC$^{•+}$ to complete the other photocatalytic cycle along with the formation of RS(O)_2OH (**J**) in the presence of water[37–40].

## Discussion

We have developed the $CO_2$-promoted photocatalytic reaction between *N*-aryl-propiolamides and sulfinates that depends on the different reaction conditions to selectively produce *tetra*- and *tri*-substituted alkenyl amides through unusual 1,4-N to C radical aryl migration. Notably, this investigation has presented $CO_2$-promoted photocatalytic radical aryl migration that enables a large variety of aryl groups to undergo fast migration from N atom to C atom, as demonstrated by more than 90 examples of the reported reaction and the efficient transformations of complex bioactive compounds. Interestingly, $CO_2$ is essential and irreplaceable for achieving the target products with excellent selectivity, thus demonstrating a new role of $CO_2$ in this reaction.

## Methods

### General procedures for the preparation of *tetra*-substituted alkenyl amides (standard conditions A)

The oven-dried Schlenk tube (38 mL) containing a stirring bar was charged with **1** (0.1 mmol), RSO_2Na (0.15 mmol, 1.5 equiv), Ir[(dFCF_3ppy)_2(bpy)]PF_6 (0.001 mmol, 1 mmol%) and DMF (2 mL). The tube was then evacuated and back-filled with $CO_2$ 3 times. The mixture was placed under a 40 W blue LED ($\lambda_{max}$ = 465 nm, 1.0 cm away from the LEDs, with cooling fan to keep the reaction temperature at 25-30°C) light source and stirred at ambient temperature for 18 h. Upon completion of the reaction, all the solvents were removed under

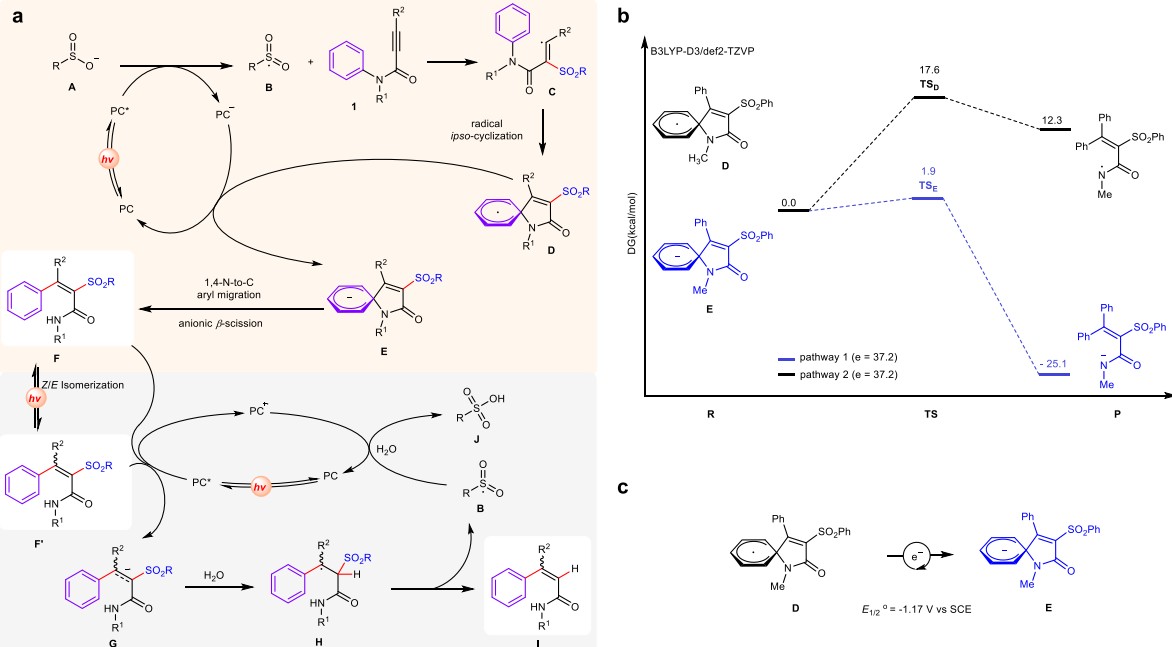

**Fig. 7 | Proposed possible mechanism. a** Outline of a possible pathway for the formation of *tetra-* and *tri-*substituted alkenyl amides. **b** Relative free energies profile for key intermediates. Calculated at the B3LYP-D3/def2-TZVP level of theory.

The Gibbs free energies (ΔG) are in kcal/mol. **R** reactant; **TS** transient state; **P** product. **c** The calculated reduction potential of intermediate **D**, details see Supporting Informtation.

reduced pressure at high temperatures. The crude reaction mixture was diluted with EtOAc (5 mL) and filtered through a short pad of Celite. The sealed tube and Celite pad were washed with an additional 25 mL of EtOAc. The filtrate was concentrated in vacuo, and crude $^1$H NMR spectrum was taken using CHCl$_2$CHCl$_2$ as an internal standard. The resulting residue was purified by flash silica gel chromatography or preparative thin layer chromatography using petroleum ether/ EtOAc (4:1–1:1) as the eluent to give the desired products.

### General procedures for the preparation of *tri-*substituted alkenyl amides (standard conditions B)

The oven-dried Schlenk tube (38 mL) containing a stirring bar was charged with **1** (0.2 mmol), PhSO$_2$Na (0.15 mmol, 1.5 equiv), Ir[(dFCF$_3$ppy)$_2$(bpy)]PF$_6$ (0.001 mmol, 1 mmol%), DMF (2 mL) and H$_2$O (0.5 mL). The tube was then evacuated and backfilled with CO$_2$ three times. The mixture was placed under a 72 W blue LED ($\lambda_{max}$ = 465 nm, 1.0 cm away from the LEDs, with a cooling fan to keep the reaction temperature at 25-30°C) light source and stirred at ambient temperature for 24 h. Upon completion of the reaction, all the solvents were removed under reduced pressure at high temperatures. The crude reaction mixture was diluted with EtOAc (5 mL) and filtered through a short pad of Celite. The sealed tube and Celite pad were washed with an additional 25 mL of EtOAc. The filtrate was concentrated in vacuo, and crude $^1$H NMR spectrum was taken using CHCl$_2$CHCl$_2$ as an internal standard. The resulting residue was purified by flash silica gel chromatography or preparative thin layer chromatography using petroleum ether/EtOAc (4:1–1:1) as the eluent to give the desired products.

### Data availability

Detailed experimental procedures, characterization data, and computational results are provided in the Supplementary Information. Crystallographic data for the structures **3**, **4**, **12a**, *Z-32*, *E-70* and *Z-70* reported in this article have been deposited at the Cambridge Crystallographic Data Center (CCDC), under deposition numbers CCDC 2282321 (**3**), 2282323 (**4**), 2380222 (**12a**), 2282339

(*Z-32*), 2283382 (*E-70*) and 2283384 (*Z-70*). These data can be obtained free of charge from The Cambridge Crystallographic Data Center via http://www.ccdc.cam.ac.uk/data_request/cif. Data supporting the findings of this manuscript are also available from the authors upon request.

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

## Acknowledgements

This work was supported by the National Key Research and Development Program of China (2018FYA0704502, W. Su), the National Natural Science Foundation of China (22271285, Y. Gao & 21931011, W. Su), Fujian Science & Technology Innovation Laboratory for Optoelectronic Information of China (2021ZZ105, W. Su).

## Author contributions

Y.G. and W.S. conceived and directed the project. G.L. performed the experiments. D.M. performed the DFT calculations. J.Z. assisted in the preparation of substrates. F.Y. performed the X-ray crystallographic structural data analysis. G.L. and Y.G. prepared the supplementary information. Y.G. wrote the manuscript.

## Competing interests

The authors declare no competing interests.
