## [Transparent Peer Review file · Nature Communications]

CO₂-Promoted Photocatalytic Aryl Migration from Nitrogen to Carbon for Switchable Transformation of N-arylpropiolamides

Corresponding Author: Professor Weiping Su

Version 0:

Reviewer comments:

Reviewer #1

(Remarks to the Author)

In this manuscript, the authors have reported a CO₂-promoted photocatalytic radical aryl migration reaction of N-arylpropiolamides with sodium sulfinates. The presence of CO₂ could greatly accelerate the visible-light-induced migration of a wide range of aryl groups from N atom to C atom via the formation of the carbon dioxide anion radical, demonstrating a novel functionality of CO₂ in catalysis. Additionally, this process utilizes the amide group as a linker to expand the radical Truce–Smiles rearrangement and selectively obtain tetra-substituted alkenyl sulfones and tri-substituted olefins under different reaction conditions. This protocol demonstrates broad substrate scope and late-stage functionalization, indicating the synthetic potential and good practicability of this reaction.

Overall, this work reveals a unique functionality of CO₂ in visible light photocatalysis that accelerates the radical Truce–Smiles rearrangement for unusual and intriguing 1,4-N to C radical aryl migration.

Therefore, this reviewer recommends this manuscript for publication in Nature Communications after some minor issues below are addressed.

Questions and suggestions:

1. In this protocol, the ortho radical addition to form the fused heterocycle would represent a competitive route. Have these by-products been detected in this reaction?
2. In the substrate scope, all products were obtained with low Z/E selectivities due to photo-induced Z/E isomerization, which is likely facilitated by an energy transfer process. Therefore, could an energy receptor be added to inhibit this process, resulting the products with high Z/E selectivities?
3. Could alkene amides serve as suitable substrates for the radical aryl migration in this reaction?
4. Although the Mechanistic studies indicate that the generated CO₂^{•-} anion radical is likely the active intermediate to promote the target reaction, it seems unlikely that transition-state E forms from CO₂^{•-} anion radical and spirocyclic radical D. A recent study by Greaney group reported a similar radical Truce–Smiles rearrangement from amides (10.1002/anie.202407979). The same spirocyclic radical intermediate directly undergoes cleavage of C-N bond to yield the final product. This reviewer wondered if the CO₂^{•-} anion radical really accelerated the rearrangement.
5. The radical ipso-cyclization is the key step for the whole transformation, and in particular in the arene dearomatization. A related recent review (ACIE 2024, e202402819) covering the ipso-cyclization might be cited.

Reviewer #2

(Remarks to the Author)

This manuscript describes a visible-light-induced radical Truce-Smiles type rearrangement of N-aryl propiolamides for the synthesis of β-aryl-α,β-unsaturated amides. An interesting finding of this study is the authors observed that the 1,4-aryl migration from nitrogen to carbon via homolytic C–N cleavage of aryl amines was accelerated in the presence of CO₂. The substrate scope of this transformation was extensively investigated and the Supporting Information was well written. Based on basic mechanistic studies, the authors proposed a possible reaction mechanism highlighting several key steps such as radical ipso-addition to aryl group, CO₂-promoted homolytic C–N bond cleavage, and a photo-induced desulfonylation in the presence of H₂O.

Actually, ref. 9, 10, 11 have reported successful examples of visible-light-induced 1,4-aryl migration from nitrogen to carbon of different kinds of N-aryl amine substrates. Regrettably, these important precedents were only briefly mentioned in the introduction part. Detailed information of these closely related reactions and fair comments should be added to put the current work in context. The corresponding reaction schemes should be included in Fig.1 as well.

One of the major concerns about this work is its mechanistic section. The acceleration of the current reaction in the presence of CO₂ is an interesting feature of this work. The authors stressed that the CO₂ radical anion interacts with spirocyclic radical intermediate D to accelerate the N-to-C radical aryl migration step. However, in this manuscript, the authors did not present any experimental results, calculations, or even evidence from literature to prove this point. Further examination is necessary to elucidate the details of this specific process (from D to F through E). More discussion about why and how CO₂ radical anion facilitates the homolytic C–N cleavage are needed to support this key conclusion.

Another issue needs to be addressed is the source of hydrogen atoms on amine product G. Although the authors ruled out the possibility of hydrogen abstraction from solvent based on deuterium experiment results, a reasonable explanation of this protonation step should be provided.

In summary, this manuscript in its current form may not meet the requirements of publishing in Nature Communications in terms of significance and potential impact in organic chemistry. Specialty journals would be a more suitable platform for the publication of this work.

Reviewer #3

(Remarks to the Author)

Gao, Su and coworkers reported radical Smiles rearrangement of aryl group from N to C, using propiolamides and arylsulfonates as precursors under CO₂ environment, leading to efficient synthesis of tri- and tetra-substituted alkenes. Indeed, N to C 1,4-migration is not well precedented, and CO₂ radical-anion mediated process (if it were generated here!) is receiving increasing attention. On a positive side, halogenated aryl group migrated well, without being reduced competitively and secondary amides served as a good substrate (9). However, there are significant limitations to this report as follows. Based on this, I have lots of reservations in recommending this manuscript for publication in Nature Comm.

- In any synthesis of alkenes (for example, olefin metathesis), control of E/Z stereochemistry is the essential issue, because it will dictate the subsequent stereoselective transformations of the products or bioactivity in a medicinal applications.

Perhaps, photocatalytic ratchet (J. Am. Chem. Soc. 2015, 137, 11254, or ref 46) may be tried to achieve E/Z control?

- Mechanism looks highly speculative. For instance, formation of E (Fig. 3) is unthinkable, because CO₂ radical anion must be nucleophilic species (polarity mismatch; if CO₂-radical anion electrophilically makes an adduct with D, it will be on the carboxyl oxygen, not nitrogen). I would like to suggest an electron transfer from CO₂-radical anion to D (not E), and 1,4-migration via an anionic pathway.

- What is the evidence that CO₂ radical anion forms? and through conPET? Known photocatalysts, 4CzIPN, and highly reducing Ir(ppy)₃ failed to give good yield (Table S1). Given that [Ir(dF(CF₃)ppy)₂bpy]⁺ have not been reported for conPET, did they measure E(0-0) and determine reducing power, according to Rehm-Weller equation? In Fig. 3f, they determined that the formate forms (3 %) in the absence of 1a, but H and C NMR data is not conclusive evidence for this. Any other evidences?

-Is SET from G to PC-radical cation possible? Cyclic voltametric analysis of G may give an answer. Or it may be an energy transfer into diradical species from G.

- In the introduction, utility of amines has been emphasized, but the authors developed synthesis of "amides". Why not reduce the claim into amides? Isn't "amine derivatives" too broad?

-Substrate scope looks a bit redundant (especially, Me, Et, Bn), but missed some key substrates: beta-alkyl or beta-unsubstituted propiolamide derivatives failed? Any N,N-diarylamides work? How about examining N,N-Ar,Ar' to see which Ar migrate faster? How about N-arylsulfonyl imides, do they work through 1,5-Ar-shift? Scope of migrating Ar group seems rather limited. Can it be extended to heteroaromatic groups, such as thiophenyl?

-Late-stage functionalization in Fig 2a: is there any meaning except to show that esterification is possible? If so, it is too trivial.

Fig 1: Yield of 3, structure of 3 was not defined.

Line 69: PC into photocatalyst

Line 88: attributed to the fact that..

Version 1:

Reviewer comments:

Reviewer #1

(Remarks to the Author)

All my concerns have been addressed properly!

Reviewer #2

(Remarks to the Author)

The author has addressed all the comments raised by the reviewers. The manuscript has been significantly improved. I support its publication in Nature Communications.

Reviewer #3

(Remarks to the Author)

In this revision, Gao, Su and coworkers responded to most of the inquiries satisfactorily. The mechanism has been more or less clarified and the scope has been expanded. However, in terms of the revised mechanism, the role of CO₂ needs further clarification, if the title has to contain the expression, "CO₂-promoted".

The authors commented that "CO₂...promote the single-electron transfer process in which the sulfonyl radical involved" and this expression is very confusing and it is not consistent with the comments that CO₂ is similar to acid catalyst. They must support the claim by a scheme. Is it Bronsted or Lewis acid catalyst?

In the revised manuscript, the authors revised the original claim (CO₂ as a redox active partner by forming CO₂ radical anion), and proposed the role as an acid. If the role of CO₂ is simply acid with "right" pK_a by forming carbonic acid (ref 46), the role of CO₂ is not so remarkable. Please note that ref 45 did not claim that CO₂ functioned as an acid.

In addition, they measured reduction potential (from 3 to radical anion) by CV to be -1.78 V. This cannot be reached with a photocatalyst having E_{ox}(PC⁺/PC^{*}) = -1.0 V. Revision is necessary.

Point-by-point response for NCOMMS-24-30181A

REVIEWER COMMENTS

Reviewer #1 (Remarks to the Author):

In this manuscript, the authors have reported a CO₂-promoted photocatalytic radical aryl migration reaction of N-aryl-propiolamides with sodium sulfinates. The presence of CO₂ could greatly accelerate the visible-light-induced migration of a wide range of aryl groups from N atom to C atom via the formation of the carbon dioxide anion radical, demonstrating a novel functionality of CO₂ in catalysis. Additionally, this process utilizes the amide group as a linker to expand the radical Truce–Smiles rearrangement and selectively obtain tetra-substituted alkenyl sulfones and tri-substituted olefins under different reaction conditions. This protocol demonstrates broad substrate scope and late-stage functionalization, indicating the synthetic potential and good practicability of this reaction.

Overall, this work reveals a unique functionality of CO₂ in visible light photocatalysis that accelerates the radical Truce–Smiles rearrangement for unusual and intriguing 1,4-N to C radical aryl migration. Therefore, this reviewer recommends this manuscript for publication in Nature Communications after some minor issues below are addressed.

Q1. In this protocol, the *ortho* radical addition to form the fused heterocycle would represent a competitive route. Have these by-products been detected in this reaction?

A: The possible fused heterocycle by-product was not detected (Fig. R1). We supposed that the formation of this oxidative by-product might not match the cycle of the photo-redox reaction conditions.

Fig. R1

Q2. In the substrate scope, all products were obtained with low *Z/E* selectivities due to photo-induced *Z/E* isomerization, which is likely facilitated by an energy transfer process. Therefore, could an energy receptor be added to inhibit this process, resulting the products with high *Z/E* selectivities?

A: Thanks for your good suggestion.

After careful investigation of photo-catalysis, solvent and light source, *tetra*- and *tri*-substituted alkenyl amides with good *Z/E* ratio were successfully achieved by using DCA as the photo-catalysis in EtOAc (Fig. R2, and these results has shown in Fig. 5a in the revised manuscript). However, it is fail to further increase the *Z/E* ratio of the *tetra*-substituted alkenyl amide for which acceptable *Z/E* ratios have been obtained in Fig.2 (**21**, **24**, **25**, **36**). To the best of our knowledge, the photocatalytic isomerization of *tetra*-substituted olefins has not been achieved until now (references 32 and 33 in the revised manuscript), presumably due to the *Z* and *E* isomers of these *tetra*-substituted olefins do not possess sufficient difference in their respective triplet

energies.

Fig. R2

Q3. Could alkene amides serve as suitable substrates for the radical aryl migration in this reaction?

A: An alkene amide has been synthesized (see annex 1 at the end of this document), but it was not a suitable substrate for the target transformation (Fig. R3) which might be due to the internal alkene amide is not an appropriate radical receptor in this radical-involved reaction. And the result has been shown in Fig. 2 in the revised manuscript (shown as “unsuccessful substrates”).

Fig. R3

Q4. Although the Mechanistic studies indicate that the generated CO₂^{•-} anion radical is likely the active intermediate to promote the target reaction, it seems unlikely that transition-state **E** forms from CO₂^{•-} anion radical and spirocyclic radical **D**. A recent study by Greaney group reported a similar radical Truce–Smiles rearrangement from amides (10.1002/anie.202407979). The same spirocyclic radical intermediate directly undergoes cleavage of C–N bond to yield the final product. This reviewer wondered if the CO₂^{•-} anion radical really accelerated the rearrangement.

A: Thanks for your good suggestions. With the assistance of DFT calculations, it was found that the transition-state **E** shown in the previous manuscript was hardly formed. Therefore, we try to make a new exploration of the mechanism:

1) Some additional examples have been conducted. a): With a slight modification of the reaction conditions, we found that sodium trifluoromethanesulfinate (CF₃SO₂Na) was also a successful precursor, furnishing the corresponding CF₃-containing *tetra*-substituted alkenyl amides in acceptable yields via C–N bond cleavage concomitant remote aryl migration (Fig. R4, and these results has shown in Fig. 4 in the revised manuscript). Notably, this transformation was proceeded successfully in the absence of CO₂. b) The expected 1,5-aryl-shifted product was obtained in an excellent yield via desulfonylation process in the presence of CO₂ when *N*-arylsulfonyl amide was employed as the substrate. On the contrary, when the reaction was conducted under N₂, the transformation was almost restrained (Fig. R5, and these results has shown in Fig. 6f in the revised manuscript).

Fig. R4

Fig. R5

As CF_3 radical triggered N-to-C aryl migration could be accessed in the absence of CO_2 (Fig. R4), we realized that role of CO_2 is not to promote the N-to-C aryl migration via the formation of $\text{CO}_2^{\cdot-}$ which is proposed in previous mechanism, but to promote the reaction in which the sulfonyl radical involved. With the assistance of DFT calculations (discuss later), the N-to-C aryl migration might proceed via a spirocyclic anionic pathway rather than a radical way. According to the references (references 42-44 in the revised manuscript), we found that the reactions involved single-electron reduction of sulfonyl-contained intermediates to generate anion intermediates required the existence of an acid. Therefore, we assumed that the role of CO_2 might be similar to that of acids, thus to promote the single-electron transfer reaction in which the sulfonyl radical involved.

2) Therefore, some control experiments have been conducted by adding acid or water to the reaction system in the absence of CO_2 (Fig. R6, and these results has shown in Fig. 6g in the revised manuscript). Expectedly, the products were also obtained, yet in low yields or with low selectivity. **Notably, compared with other acids, CO_2 is able to synthesize tetra-substituted alkenyl amides and tri-substituted alkenyl amides with excellent selectivity in generally good yields.** We assume that the addition of a protonic acid could not only promote the formation of tetra-substituted alkenyl amides, but also facilitate the generation of tri-substituted alkenyl amides (Fig. 3 and Fig. 6d in the revised manuscript), thus yielding the bad selectivity of products when an acid was added. As a result, we believe that CO_2 does not directly participate in the reaction, but to promote the single-electron transfer process in which the sulfonyl radical involved.

Fig. R6

3) Moreover, DFT calculations have been performed in order to provide the energy landscape of the key intermediates (Fig. R7, and some of these results has shown in Fig. 7b in the revised manuscript). It was found that the cleavage of the C–N bond via a spirocyclic anionic pathway was quite facile, associated with a small energetic barrier at the transition state **TS_E** (1.9 kcal mol⁻¹, blue line). By comparison, the homolytic cleavage of the C–N bond via a radical process presented significant challenges, with a high energetic barrier at the transition state **TS_D** (17.6 kcal mol⁻¹, black line). A similar homolytic dissociation mechanism was reported in previous experimental work (10.1002/anie.202407979). Here, we calculated the reaction process at the same computational level for comparison. Notably, the homolytic cleavage of the C–N bond reported in the work referenced also exhibits a high energetic barrier (15.1 kcal mol⁻¹, orange line), and we do not comment further on the details of that work here, but we have cited the work as reference 14 in the revised manuscript. Notably, the migratory arenes shown in this work were limited to *ortho*-substituted benzene derivatives.

Fig. R7

Additional, after DFT calculations, it was shown that the single-electron reduction from **D** to **E** was feasible ($E_{1/2}^{\circ} = -1.17$ V vs SCE; Fig. R8, calculation details see SI) in the presence of Ir[(dFCF₃ppy)₂(bpy)]PF₆ [$E_{1/2}^{\text{red}}(\text{PC}/\text{PC}^{\bullet-}) = -1.37$ V vs SCE for Ir[(dFCF₃ppy)₂(bpy)]PF₆].

Fig. R8

In conclusion, the cleavage of C–N bond is via photocatalytic radical-polar crossover anion pathway rather than a radical way (details see Fig. 7a in the revised manuscript). We believe that CO₂ does not directly participate in the reaction, but to promote the single-electron transfer process in which the sulfonyl radical involved. Notably, CO₂ is essential and irreplaceable for achieving the target products with excellent selectivity in good yields.

Although the revised mechanism does not involve the formation of CO₂^{•-}, yet the revised mechanism is more convincing through a series of control experiments and DFT calculations. We hope these changes are acceptable.

Q5. The radical ipso-cyclization is the key step for the whole transformation, and in particular in the arene dearomatization. A related recent review (ACIE 2024, e202402819) covering the ipso-cyclization might be cited.

A: This review has been cited in reference 49 in the revised manuscript.

Reviewer #2 (Remarks to the Author):

This manuscript describes a visible-light-induced radical Truce-Smiles type rearrangement of N-aryl propiolamides for the synthesis of β -aryl- α,β -unsaturated amides. An interesting finding of this study is the authors observed that the 1,4-aryl migration from nitrogen to carbon via homolytic C–N cleavage of aryl amines was accelerated in the presence of CO₂. The substrate scope of this transformation was extensively investigated and the Supporting Information was well written. Based on basic mechanistic studies, the authors proposed a possible reaction mechanism highlighting several key steps such as radical ipso-addition to aryl group, CO₂-promoted homolytic C–N bond cleavage, and a photo-induced desulfonylation in the presence of H₂O.

Q1. Actually, ref. 9, 10, 11 have reported successful examples of visible-light-induced 1,4-aryl migration from nitrogen to carbon of different kinds of N-aryl amine substrates. Regrettably, these important precedents were only briefly mentioned in the introduction part. Detailed information of these closely related reactions and fair comments should be added to put the current work in context. The corresponding reaction schemes should be included in Fig.1 as well.

A: Thanks for your suggestion, the introduction has been rewritten and these closely related reactions have been detailed commented. We hope the changes are acceptable.

Q2. One of the major concerns about this work is its mechanistic section. The acceleration of the current reaction in the presence of CO₂ is an interesting feature of this work. The authors stressed that the CO₂ radical anion interacts with spirocyclic radical intermediate **D** to accelerate the N-to-C radical aryl migration step. However, in this manuscript, the authors did not present any experimental results, calculations, or even evidence from literature to prove this point. Further examination is necessary to elucidate the details of this specific process (from **D** to **F** through **E**). More discussion about why and how CO₂ radical anion facilitates the homolytic C–N cleavage are needed to support this key conclusion.

A: Thanks for your suggestions. The mechanism was amended with the assistance of DFT calculations (Fig. 7b, Fig. 7c), and the role of CO₂ was confirmed through additional examples (Fig.4 and Fig. 6f) and more control experiments (Fig. 6g).

Firstly, DFT calculations have been performed in order to provide the energy landscape of the key intermediates. It was found that the transition-state **E** shown in the previous manuscript was hardly formed. And the cleavage of the C–N bond via a spirocyclic anionic pathway was quite facile, associated with a small energetic barrier at the transition state **TS_E** (1.9 kcal mol⁻¹, blue line). By comparison, the homolytic cleavage of the C–N bond via a radical process presented significantly challenges, with a high energetic barrier at the transition state **TS_D** (17.6 kcal mol⁻¹, black line). (Fig. R9, Fig. 7b in the revised manuscript)

Fig. R9

Additional, after DFT calculations, it was shown that the single-electron reduction from **D** to **E** was feasible ($E_{1/2}^{\circ} = -1.17$ V vs SCE; Fig. R10, calculation details see SI) in the presence of Ir[(dFCF₃ppy)₂(bpy)]PF₆ [$E_{1/2}^{\text{red}}(\text{PC}/\text{PC}^{\bullet-}) = -1.37$ V vs SCE for Ir[(dFCF₃ppy)₂(bpy)]PF₆]. Thus might rule out the possibility that an electron transfer from CO₂-radical anion to **D**.

Fig. R10

Therefore, to explore the role of CO₂ in this reaction, more experiments have been conducted including additional examples and control experiments.

Some additional examples have been conducted. a): With a slight modification of the reaction conditions, we found that sodium trifluoromethanesulfinate (CF₃SO₂Na) was also a successful precursor, furnishing the corresponding CF₃-containing *tetra*-substituted olefins in acceptable yields via C–N bond cleavage concomitant remote aryl migration (Fig. R11, and these results has shown in Fig. 4 in the revised manuscript). Notably, this transformation was proceeded successfully in the absence of CO₂. b) The expected 1,5-aryl-shifted product was obtained in an excellent yield via desulfonylation process in the presence of CO₂ when *N*-arylsulfonyl amide was employed as the substrate. On the contrary, when the reaction was conducted under N₂, the transformation was almost restrained (Fig. R12, and these results has shown in Fig. 6f in the revised manuscript).

Fig. R11

Fig. R12

As CF₃ radical triggered N-to-C aryl migration also could be accessed in the absence of CO₂ (Fig. R11), we realized that role of CO₂ is not to promote the N-to-C aryl migration via the formation of CO₂^{•-} which is proposed in previous mechanism, but to promote the reaction in which the sulfonyl radical involved. DFT calculations have shown that the N-to-C aryl migration proceed via a spirocyclic anionic pathway rather than a radical way. According to the references (references 42-44 in the revised manuscript), we found that the reactions involved single-electron reduction of sulfonyl-contained intermediates to generate anion intermediates required the existence of an acid. Therefore, we assumed that the role of CO₂ might be similar to that of acids, thus to promote the single-electron reduction of sulfonyl radical.

As a result, some control experiments have been conducted by adding acid or water to the reaction system in the absence of CO₂ (Fig. R13, and these results has shown in Fig. 6g in the revised manuscript). Expectedly, the products were also obtained, yet in low yields or with low selectivity. Notably, compared with other acids, CO₂ is able to synthesize *tetra*-substituted olefins and *tri*-substituted olefins with excellent selectivity in generally good yields. We assume that the addition of a protonic acid not only could promote the formation of *tetra*-substituted olefins, but also facilitate the generation of *tri*-substituted olefins (Fig. 3 and Fig. 6c in the revised manuscript), thus yielding the bad selectivity of products when an acid was added. As a result, we believe that CO₂ does not directly participate in the reaction, but to promote the single-electron process in which sulfonyl-contained intermediates involved.

Fig. R13

In conclusion, the cleavage of C–N bond is via an anionic pathway rather than a radical way. We believe that CO₂ does not directly participate in the reaction, but to promote the single-electron transfer process in which sulfonyl-contained intermediates (details see Fig. 7a in the revised manuscript). Notably, CO₂ is essential and irreplaceable for achieving the target products with excellent selectivity in good yields.

Although the revised mechanism does not involve the formation of CO₂^{•-}, yet the revised mechanism is more convincing through a series of control experiments and DFT calculations. We hope these changes are acceptable.

Q3. Another issue needs to be addressed is the source of hydrogen atoms on amine product **G**. Although the authors ruled out the possibility of hydrogen abstraction from solvent based on deuterium experiment results, a reasonable explanation of this protonation step should be provided.

A: In fact, most of the sodium sulfonates were not commercially available, but were synthesized in our lab. During the experiment, the sodium sulfonates were recrystallized from water. Although the products have been dried, we assumed that the residual crystal water (also for commercially available sodium sulfonates) might account for the source of hydrogen atoms in the products. And we have explained it in the revised manuscript. *“The residual crystal water from the sodium sulfonates might account for the source of hydrogen atoms in the products **F**.”*

Reviewer #3 (Remarks to the Author):

Gao, Su and coworkers reported radical Smiles rearrangement of aryl group from N to C, using propiolamides and arylsulfonates as precursors under CO₂ environment, leading to efficient synthesis of tri- and tetra-substituted alkenes. Indeed, N to C 1,4-migration is not well precedented, and CO₂ radical-anion mediated process (if it were generated here!) is receiving increasing attention. On a positive side, halogenated aryl group migrated well, without being reduced competitively and secondary amides served as a good substrate. However, there are significant limitations to this report as follows. Based on this, I have lots of reservations in recommending this manuscript for publication in Nature Comm.

Thanks for your good suggestions sincerely, as the mechanism and substrate scope of the transformation have been improved to a large extent after we revised the manuscript according to your suggestions!

Q1. In any synthesis of alkenes (for example, olefin metathesis), control of E/Z stereochemistry is the essential issue, because it will dictate the subsequent stereoselective transformations of the products or bioactivity in a medicinal applications. Perhaps, photocatalytic ratchet (J. Am. Chem. Soc. 2015, 137, 11254, or ref 46) may be tried to achieve E/Z control?

A: After careful investigation of photo-catalysis, solvent and light source, *tetra*- and *tri*-substituted alkenyl amides with good Z/E ratio were successfully achieved by using DCA as the photo-catalysis in EtOAc (Fig. R14, and these results has shown in Fig. 5a in the revised manuscript). However, it is fail to further increase the Z/E ratio of the *tetra*-substituted alkenyl amide for which acceptable Z/E ratios have been obtained in Fig.2 (21, 24, 25, 36). To the best of our knowledge, the photocatalytic isomerization of *tetra*-substituted olefins has not been achieved until now (references 32 and 33 in the revised manuscript), presumably due to the Z and E isomers of these *tetra*-substituted olefins do not possess sufficient difference in their respective triplet energies.

Fig. R14

Q2. Mechanism looks highly speculative. For instance, formation of **E** (Fig. 3) is unthinkable, because CO₂ radical anion must be nucleophilic species (polarity mismatch; if CO₂-radical anion electrophilically makes an adduct with **D**, it will be on the carboxyl oxygen, not nitrogen). I would like to suggest an electron transfer from CO₂-radical anion to **D** (not **E**), and 1,4-migration via an anionic pathway.

A: Thanks for your good suggestions. DFT calculations have been performed in order to provide the energy landscape of the key intermediates. Firstly, it was found that the transition-state **E** shown in the previous manuscript was hardly formed thus rule out the mechanism proposed before.

As you might expect (Fig. R15, Fig. 7b in the revised manuscript), the cleavage of the C–N bond via a spirocyclic anionic pathway was quite facile, associated with a small energetic barrier at the transition state **TS_E** (1.9 kcal mol⁻¹, blue line). By comparison, the homolytic cleavage of the C–N bond via a radical process presented significantly challenges, with a high energetic barrier at the transition state **TS_D** (17.6 kcal mol⁻¹, black line).

Fig. R15

Additional, after DFT calculations, it was shown that the single-electron reduction from **D** to **E** was feasible ($E_{1/2}^{\circ} = -1.17$ V vs SCE; Fig. R16, calculation details see SI) in the presence of Ir[(dFCF₃ppy)₂(bpy)]PF₆ [$E_{1/2}^{\text{red}}(\text{PC}/\text{PC}^{\bullet-}) = -1.37$ V vs SCE for Ir[(dFCF₃ppy)₂(bpy)]PF₆]. Thus might rule out the possibility that an electron transfer from CO₂-radical anion to **D**.

Fig. R16

To explore the role of CO₂ in this reaction, more experiments have been conducted including additional examples and control experiments.

Some additional examples have been conducted. a): With a slight modification of the reaction conditions, we found that sodium trifluoromethanesulfinate (CF₃SO₂Na) was also a successful precursor, furnishing the corresponding CF₃-containing *tetra*-substituted olefins in acceptable yields via C–N bond cleavage concomitant remote aryl migration (Fig. R17, and these results has shown in Fig. 4 in the revised manuscript). Notably, this transformation was proceeded successfully in the absence of CO₂. b) The expected 1,5-aryl-shifted product was obtained in an excellent yield via desulfonylation process in the presence of CO₂ when *N*-arylsulfonyl amide was employed as the substrate. On the contrary, when the reaction was conducted under N₂, the transformation was almost restrained (Fig. R18, and these results has shown in Fig. 6f in the revised manuscript).

Fig. R17

Fig. R18

As CF_3 radical triggered N-to-C aryl migration also could be accessed in the absence of CO_2 (Fig. R17), we realized that role of CO_2 is not to promote the N-to-C aryl migration via the formation of $\text{CO}_2^{\cdot-}$ which is proposed in previous mechanism, but to promote the reaction in which the sulfonyl radical involved. It was mentioned above that the N-to-C aryl migration proceed via a spirocyclic anionic pathway rather than a radical way. According to the references (ref. 42-44 in the revised manuscript), we found that the reactions involved single-electron reduction of sulfonyl-contained intermediates to generate anion intermediates required the existence of an acid. Therefore, we assumed that the role of CO_2 might be similar to that of acids, thus to promote the single-electron transfer reaction in which sulfonyl radical involved.

As a result, some control experiments have been conducted by adding acid or water to the reaction system in the absence of CO_2 (Fig. R19, and these results has shown in Fig. 6g in the revised manuscript). Expectedly, the products were also obtained, yet in low yields or with low selectivity. *Notably, compared with other acids, CO_2 is able to synthesize tetra-substituted olefins and tri-substituted olefins with excellent selectivity in generally good yields.* We assume that the addition of a protonic acid could not only promote the formation of *tetra*-substituted olefins, but also facilitate the generation of *tri*-substituted olefins (Fig. 3 and Fig. 6c in the revised manuscript), thus yielding the bad selectivity of products when an acid was added. As a result, we believe that CO_2 does not directly participate in the reaction, but to promote the single-electron transfer process in which sulfonyl-contained intermediates.

Fig. R19

In conclusion, the cleavage of C–N bond is via an anionic pathway rather a radical way. We believe that CO₂ does not directly participate in the reaction, but to promote the single-electron transfer process in which sulfonyl-contained intermediates (details see Fig. 7a in the revised manuscript).

Although the revised mechanism does not involve the formation of CO₂^{•-}, yet the revised mechanism is more convincing through a series of control experiments and DFT calculations. Notably, CO₂ is essential and irreplaceable for achieving the target products with excellent selectivity in good yields. We hope these changes are acceptable.

Q3. What is the evidence that CO₂ radical anion forms? and through conPET? Known photocatalysts, 4CzIPN, and highly reducing Ir(ppy)₃ failed to give good yield (Table S1). Given that [Ir(dF(CF₃)ppy)₂bpy]⁺ have not been reported for conPET, did they measure E(0-0) and determine reducing power, according to Rehm-Weller equation? In Fig. 3f, they determined that the formate forms (3 %) in the absence of **1a**, but H and C NMR data is not conclusive evidence for this. Any other evidences?

A: As the revised mechanism excluded the formation of CO₂ radical anion, experiments to verify the generation of CO₂ radical anion were not carried out further.

Q4. Is SET from **G** to PC-radical cation possible? Cyclic voltametric analysis of **G** may give an answer. Or it may be an energy transfer into diradical species from **G**.

A: Thanks for your suggestion. **G** refers to *tetra*-substituted alkenyl amide in the previous manuscript. Therefore, cyclic voltammetry (CV) experiments of *tetra*-substituted alkenyl amide **3** was conducted, and it showed a significant reduction peak ($E_{\text{red}} = -1.78$ V vs SCE, Fig. 20, and these results has shown in Fig. S8 in the revised SI), suggesting the possibility that compound **3** might be reduced by PC [$E_{\text{ox}}(\text{PC}^*/\text{PC}^{++}) = -1.00$ V vs SCE for Ir[(dFCF₃ppy)₂(bpy)]PF₆]. To further confirm this possibility, EtOH was used as the hydrogen source instead of H₂O in this desulfurization reaction, oxidized compound **99** was detected by GC-MS, indicating that this desulfurization might through a SET process rather an energy transfer (EnT) process (Fig. R21, and these results has shown in Fig. 6d in the revised manuscript).

Fig. R20

Fig. R21

Q5. In the introduction, utility of amines has been emphasized, but the authors developed synthesis of “amides”. Why not reduce the claim into amides? Isn’t “amine derivatives” too broad?

A: The introduction has been rewritten, and we narrowed down the topic from amines to amides. We hope the changes are acceptable.

Q6. Substrate scope looks a bit redundant (especially, Me, Et, Bn), but missed some key substrates: 1) beta-alkyl or beta-unsubstituted propiolamide derivatives failed? 2) Any N,N-diarylamides work? How about examining N,N-Ar,Ar’ to see which Ar migrate faster? 3) How about N-arylsulfonyl imides, do they work through 1,5-Ar-shift? 4) Scope of migrating Ar group seems rather limited. Can it be extended to heteroaromatic groups, such as thiophenyl?

A: 1) beta-alkyl or beta-unsubstituted propiolamide derivatives failed?

β -Alkyl propiolamide and β -unsubstituted propiolamide derivatives have been synthesized (see annexs 2 and 3 at the end of this document). β -Alkyl propiolamide was not compatible with this transformation, possibly because the alkyl substituted alkenyl radical intermediate is unable to facilitate the following radical *ipso*-addition. And sulfonyl radical was more likely to attack to the

terminal position of alkyne when β -unsubstituted propiolamide was employed as the substrate (Fig. R22). And the result has been shown in Fig. 2 in the revised manuscript (shown as “unsuccessful substrates”).

Figure R22

2) Any N,N -diarylamides work? How about examining N,N -Ar,Ar' to see which Ar migrate faster?

N,N -diarylamides have been synthesized and examined. N,N -diaryl propiolamides were also compatible with the migratory transformation and the aryl migratory ability of different benzyl tethered to N atom was also examined (Fig. R23, and these results has shown in Fig. 2 in the revised manuscript). Both benzyl and 4-fluorobenzyl groups were able to serve as migratory arenes, but the relatively more electron-poor 4-fluorophenyl group was preferred, resulting in a regioselective ratio (rr) of 3:1 (**11**). Another pair comprising 4-methoxybenzyl and benzyl groups showed lower selectivity to produce **12** with rr = 1.4:1.

Figure R23

3) How about N -arylsulfonyl imides, do they work through 1,5-Ar-shift?

N -arylsulfonyl amide was synthesized (see annex 4 at the end of this document). The expected 1,5-aryl-shifted product was obtained in an excellent yield (Fig. R24, and these results has shown in Fig. 6f in the revised manuscript) via desulfonylation process under 1 atm of CO_2 . On the contrary, when the reaction was conducted under N_2 , the transformation was almost restrained, and the starting material and protonated by-product were detected in 42% and 20% yields,

respectively.

Figure R24

4) Scope of migrating Ar group seems rather limited. Can it be extended to heteroaromatic groups, such as thiophenyl?

Substrates bearing benzo-fused migrating groups (such as carbazolyl, indolyl and benzo-1,3-dioxol) were synthesized (see annexs 5 and 6 at the end of this document), and all of them were tested under the standard reaction conditions. Benzo-1,3-dioxol was successfully migrated in a good yield, while the carbazolyl and indolyl produced the target products in low yields, possibly because carbazolyl and indolyl are more difficult to dearomatize.

Additional, substrates bearing a thiophenyl as the migrating group was also synthesized, and the corresponding product was obtained in an acceptable yield. [Fig. R25, and these results has shown in Fig. 2 (products **40** and **42**) in the revised manuscript]

Figure R25

Q7. Late-stage functionalization in Fig 2a: is there any meaning except to show that esterification is possible? If so, it is too trivial.

A: Sodium sulfonates from Valdecocixib and Sildenafil were prepared, and both of them were

compatible with the target transformation in acceptable yields [Fig. R26, and these results has shown in Fig. 2 (products **62** and **63**) in the revised manuscript]. We hope these examples would enrich the late-stage functionalization.

Figure R26

Q8. Fig 1: Yield of **3**, structure of **3** was not defined.

Line 69: PC into photocatalyst

Line 88: attributed to the fact that.

A: The structure of **3** has been defined in Fig. 1. “PC” has been changed to “photocatalyst”, and “attributed to” has been changed to “attributed to the fact that” in the revised manuscript.

Annex:

annex 1

mixture of rotamers

annex 2

annex 3

annex 4

mixture of rotamers

annex 5

mixture of rotamers

annex 6

Point-by-point response for NCOMMS-24-30181A

Reviewer #1 (Remarks to the Author):

All my concerns have been addressed properly!

Reviewer #2 (Remarks to the Author):

The author has addressed all the comments raised by the reviewers. The manuscript has been significantly improved. I support its publication in Nature Communications.

Reviewer #3 (Remarks to the Author):

Q: In this revision, Gao, Su and coworkers responded to most of the inquiries satisfactorily. The mechanism has been more or less clarified and the scope has been expanded. However, in terms of the revised mechanism, the role for CO₂ needs further clarification, if the title has to contain the expression, "CO₂-promoted". The authors commented that "CO₂...promote the single-electron transfer process in which the sulfonyl radical involved" and this expression is very confusing and it is not consistent with the comments that CO₂ is similar to acid catalyst. They must support the claim by a scheme. Is it Bronsted or Lewis acid catalyst?

A: 1) The expression "CO₂...promote the single-electron transfer process in which the sulfonyl radical involved" has been deleted in the final manuscript.

2) Some more control experiments were performed by adding Bronsted or Lewis acid to the reaction system in the absence of CO₂ to explore the role of CO₂ in this reaction. As the results shown in Fig. 6g, the products could also be obtained in the presence of Bronsted acid, yet in low yields or with low selectivity. Oppositely, Lewis acid, such as ZnF₂, could not access satisfactory results. As a result, we assumed that the role of CO₂ might be similar to that of Bronsted acids. (These results have been revised in the final manuscript.)

Figure 6g in the manuscript

Q: In the revised manuscript, the authors revised the original claim, and proposed the role as an acid. If the role of CO₂ is simply acid with "right" pKa by forming carbonic acid (ref 46), the role of CO₂ is not so remarkable. Please note that ref 45 did not claim that CO₂

functioned as an acid.

A: Although in the revised manuscript, CO₂ is proposed to act as an acid, yet compared with other acids, CO₂ is able to synthesize *tetra*- and *tri*-substituted alkenyl amides with excellent selectivity in generally good yields, demonstrating the essential and irreplaceable role of CO₂ for achieving the target products with excellent selectivity. In conclusion, although CO₂ does not play a remarkable role in the mechanism issue, it is essential for the synthesis of specific products. We suppose that this issue will appeal to readers. And ref 45 has been revised.

Q: In addition, they measured reduction potential (from **3** to radical anion) by CV to be -1.78 V. This cannot be reached with a photocatalyst having $E_{\text{ox}}(\text{PC}^+/\text{PC}^*) = -1.0$ V. Revision is necessary.

A: The results shown in Fig. 6d in the final manuscript indicated that a SET process was involved in the desulfurization reaction.

Figure 6d in the manuscript

We have measured the reduction potential of product **3** by CV, yet the measured potential cannot be reached by the PC to some extent. To confirm the result, we calculate the reduction potential of product **3** and the result is -1.18 V (calculation details see Supporting Information, Page S46-S48) which is more matched with the potential of PC. Therefore, the potential of product **3** has been revised in the final manuscript, and we hope the correction was acceptable.